# A taxonomy for detecting and preventing temporal data leakage in machine learning-based build prediction: A dual-platform empirical validation

**Lalit Narayan Mishra**[1]*, **Amit Rangari**[2], **Sandesh Nagrare**[3], **Saroj Kumar Nayak**[4]

1 Lowe's Companies, Inc., Charlotte, North Carolina, United States of America, 2 JPMorgan Chase & Co, Atlanta, Georgia, United States of America, 3 Digital Remedy, New York, New York, United States of America, 4 Cognizant Technology Solutions, Charlotte, North Carolina, United States of America

* Lnm8910@gmail.com

## Abstract

Modern software development relies on automated build systems that compile and test code whenever developers make changes. Predicting whether these builds will succeed or fail before execution could save computational resources and developer time. However, many machine learning models for build prediction suffer from temporal data leakage, a methodological flaw where the model inadvertently uses information that would only be available after the build completes, producing artificially inflated accuracy that fails in real-world deployment. This study develops a three-type taxonomy to systematically identify and prevent such leakage: (1) Direct Outcome Encoding (using the build result itself as a feature), (2) Execution-Dependent Metrics (information generated during build execution), and (3) Future Information Leakage (using data from chronologically later builds). Applying this taxonomy reveals that prior studies reporting 95–99% accuracy likely used contaminated features, while realistic accuracy is substantially lower. The methodology is validated on 175,706 builds from two open-source CI/CD platforms spanning 10 years: TravisTorrent (100,000 builds, 2013–2017) and GHALogs (75,706 workflows, 2023). Removing leaky features reduces accuracy by 15.07 percentage points on TravisTorrent (97.8% to 82.73%) but only 0.48 points on GHALogs (83.77% to 83.30%), revealing that modern GitHub Actions' tight integration with repositories enables accurate prediction from static project metadata alone. Using only legitimately available pre-build features, Random Forest classifiers achieve 82.73% (TravisTorrent) and 83.30% (GHALogs) accuracy, sufficient for practical deployment. Surprisingly, project maturity and build history prove more predictive than code complexity metrics, suggesting organizational factors outweigh code quality. The models generalize across programming languages (Java, Ruby, Python, JavaScript) with minimal performance variation.

**Data availability statement:** The data and replication package remain publicly available via Zenodo (DOI: 10.5281/zenodo.17745286), alongside the original TravisTorrent (DOI: 10.5281/zenodo.1254890) and GHALogs (DOI: 10.5281/zenodo.10154920) datasets.

**Funding:** The author(s) received no specific funding for this work.

**Competing interests:** The authors have declared that no competing interests exist. LNM is employed by Lowe's Companies, Inc., AR by JPMorgan Chase & Co., SN by Digital Remedy, and SKN by Cognizant Technology Solutions; these affiliations did not influence the research. The specific roles of these authors are articulated in the 'author contributions' section. There are no patents, products in development or marketed products associated with this research to declare. This does not alter our adherence to PLOS ONE policies on sharing data and materials.

Open-source tools for detecting temporal leakage in any software prediction task are provided.

---

## Introduction

### Background and motivation

Continuous Integration/Continuous Deployment (CI/CD) systems automatically compile code, run tests, and verify software correctness whenever developers commit changes [1]. Build failures occur in 20–40% of cases [2,3], blocking releases, wasting computational resources, and requiring 30–60 minutes of developer time to diagnose [4]. Predicting build outcomes before execution could enable intelligent resource allocation and instant developer feedback, yet machine learning studies reporting 95–99% prediction accuracy [5] remain largely undeployed in practice.

The gap between reported accuracy and deployment viability stems from temporal data leakage: models inadvertently trained on features available only after the predicted event occurs. In build prediction, features such as `tr_status` (build outcome itself), `tr_duration` (execution time), and `tr_log_tests_failed` (post-execution test counts) from the widely-used TravisTorrent dataset [6] provide perfect retrospective discrimination but are definitionally unavailable at prediction time. Studies restricting to genuinely pre-build features report 75–84% accuracy [2,3], a 15–20 percentage point gap quantifying the leakage impact.

### The methodological gap

Kaufman et al. [7] established a foundational leakage taxonomy for general machine learning, and Kapoor and Narayanan [8] demonstrated that leakage contributes to a reproducibility crisis across 17 scientific fields (294 affected papers). However, software engineering datasets exhibit unique temporal leakage risks due to the sequential nature of development activities, manifesting in defect prediction [9], test selection [10], and code review automation [11]. Despite growing awareness, no systematic methodology exists for detecting and preventing temporal leakage specifically in software engineering prediction tasks. Prior work identifies individual leakage instances but lacks generalizable frameworks with detection rules and prevention guidelines applicable across domains.

### Approach

This study develops a three-type temporal leakage taxonomy (Direct Outcome Encoding, Execution-Dependent Metrics, Future Information Leakage) and validates it on 175,706 builds from two independent CI/CD platforms spanning 10 years: TravisTorrent [6] (100,000 Travis CI builds, 2013–2017) and GHALogs [12] (75,706 GitHub Actions workflows, 2023). This dual-platform design assesses methodology robustness across major infrastructure shifts. Dataset details are provided in Materials and Methods.

## Research contributions

This research presents five main contributions advancing CI/CD build prediction methodology through dual-platform empirical validation:

1. **Three-Type Temporal Leakage Taxonomy**: A systematic taxonomy categorizing leaky features as Direct Outcome Encoding, Execution-Dependent Metrics, or Future Information Leakage. Cross-platform validation reveals divergent leakage impact: 15.07pp inflation on TravisTorrent versus 0.48pp on GHALogs, demonstrating platform-dependent metadata predictiveness.

2. **Dual-Dataset Validation (175,706 Builds, 11 Years)**: Leakage-free Random Forest achieves 82.73% (TravisTorrent, 31 features) and 83.30% (GHALogs, 29 features) accuracy, representing realistic deployment performance contrasting with inflated 95–99% claims from leakage-contaminated studies.

3. **Project Maturity Dominates Code Metrics**: Project context accounts for 49.8% of feature importance versus 7.7% for code metrics (6.5:1 ratio), challenging code-centric software engineering paradigms and demonstrating that organizational factors predict build outcomes more reliably than code characteristics.

4. **Cross-Language Generalization**: Minimal performance variation across Java, Ruby, Python, and JavaScript (range: 3.38pp), validating single multi-language model deployment without per-language retraining.

5. **Open-Source Leakage Detection Toolkit**: Complete replication package (Zenodo DOI: 10.5281/zenodo.17745286) with datasets, training code, and automated leakage detection scripts applicable to defect prediction, test selection, and code review prioritization.

## Related work context

This work builds upon three fundamental research areas: machine learning applications in software engineering, CI/CD analytics and build prediction, and temporal data integrity in predictive modeling.

**Machine learning for software engineering.** Machine learning has advanced software engineering tasks including defect prediction, automated testing, and development analytics [13–15]. Grillmeyer et al. [16] introduced measures for quantifying data leakage in failure prediction tasks, demonstrating that leakage-prone splitting techniques significantly overestimate model performance, directly applicable to CI/CD build prediction. Deep learning approaches (LSTM [17], Transformers [18], GNNs [19]) have shown promise but require extensive data and offer limited interpretability compared to ensemble methods.

Ensemble learning methods achieve state-of-the-art results for tabular software metrics [9,20–22]. Random Forest and Gradient Boosting provide competitive accuracy with superior explainability and lower computational cost than deep learning [23,24], motivating their selection for this study alongside Logistic Regression and Decision Tree baselines.

**CI/CD analytics and build prediction.** Build failure characterization studies report 20–40% failure rates in CI/CD systems [2,3], with causes including compilation errors, flaky tests, and dependency problems [25,26]. The TravisTorrent dataset [6], the primary benchmark for build prediction research (40+ published papers, 2.64 million builds), has been analyzed extensively. The DL-CIBuild study [5] achieved over 95% accuracy using LSTM networks, but these results stem from including post-execution features (build status, test counts, execution duration) unavailable at prediction time. Sun et al. [27] introduced RavenBuild, a context-aware approach achieving 50% F1 improvement over baselines in industrial settings. Seow et al. [28] demonstrated Random Forest viability for build time prediction on enterprise data. Studies restricting to pre-build features achieve 75–84% accuracy [2,3], aligning with the leakage-free results presented here.

Modern CI/CD platforms have evolved substantially since TravisTorrent (2013–2017). GitHub Actions research reveals workflow optimization opportunities [29,30] and platform-specific patterns [31,32]. This platform evolution motivates the dual-dataset validation strategy employed here.

**Software development metrics and performance indicators.** Software Development Lifecycle (SDLC) metrics quantify development activities across project phases. Code complexity [33], code churn [34,35], and commit patterns [36] inform build prediction features. The DORA metrics framework [37,38] demonstrates that organizational maturity (deployment frequency, change failure rate) correlates strongly with software delivery performance, motivating the inclusion of project maturity and build frequency features in prediction models.

Rigorous evaluation of prediction models requires metrics beyond accuracy for imbalanced data. Matthews Correlation Coefficient (MCC) provides balanced treatment of all confusion matrix elements [39], precision-recall analysis avoids misleading conclusions from accuracy alone [40], and statistical testing (Wilcoxon, Bonferroni correction, effect sizes) ensures reliable model comparisons [41,42]. Forward chaining cross-validation preserves temporal ordering for time-dependent data [43].

## Materials and methods

This section describes the experimental methodology for CI/CD (Continuous Integration/Continuous Deployment) build prediction, including dataset processing, temporal data leakage prevention, feature engineering, and machine learning model development. Fig 1 provides an overview of the six-layer architecture that processes builds from raw data through leakage prevention to production deployment. The approach evaluates whether software metrics measurable before build

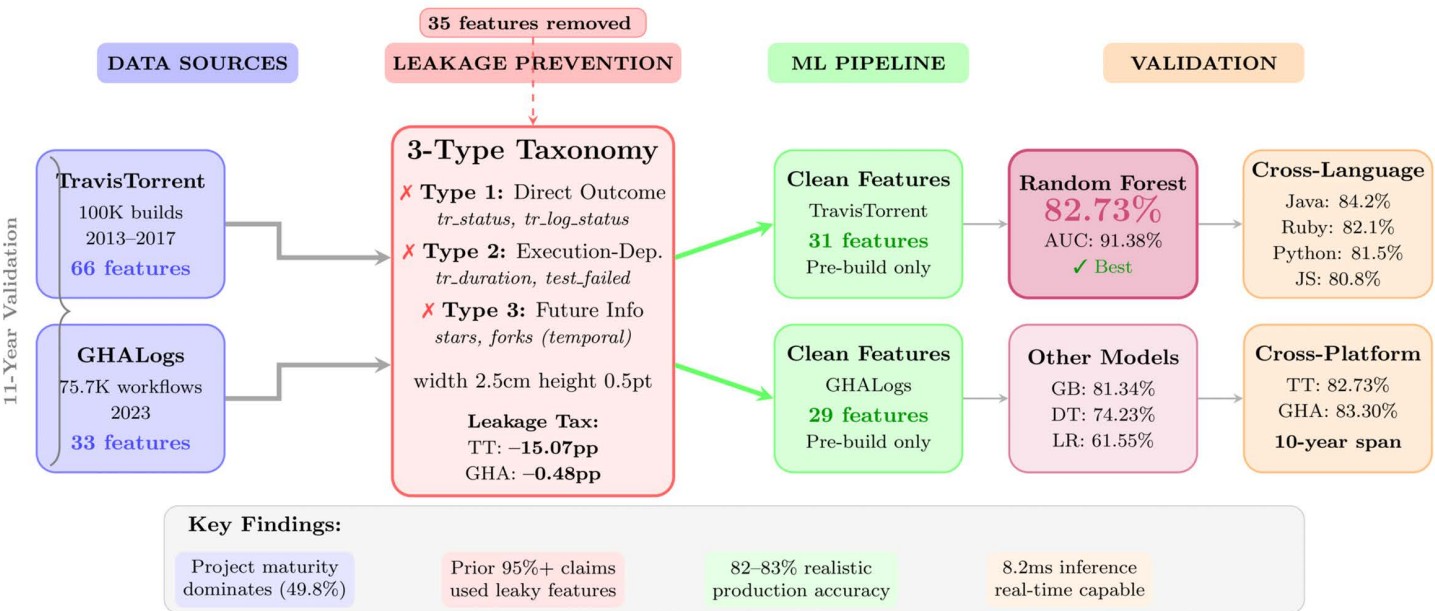

**Fig 1. CI/CD build prediction system architecture with temporal data leakage prevention. Description:** Six-layer pipeline processing 175,706 builds from TravisTorrent (100,000 builds, 2013–2017) and GHALogs (75,706 workflows, 2023). The three-type leakage taxonomy filters features from 66 to 31 (TravisTorrent) and 33–29 (GHALogs). Random Forest achieves 82.73–83.30% accuracy using only pre-build features, with project maturity (49.8% importance) dominating code metrics (7.7%).

execution can accurately predict build outcomes, following established empirical software engineering guidelines for rigor and reproducibility [44].

## Datasets

The temporal leakage taxonomy and prediction methodology are validated on two independent datasets spanning 10 years of CI/CD platform evolution.

**TravisTorrent dataset.** TravisTorrent [6] synthesizes Travis CI build logs with GitHub repository metadata from over 1,000 open-source projects [45,46], encompassing 2.64 million builds (January 2013 to December 2017) across Java (402 projects), Ruby (443), Python (218), and JavaScript (220) [47]. Each build record contains 66 features spanning project context, build context, commit metrics, code complexity, and test structure, with build outcome (passed/failed) as the target variable. The dataset is publicly available via Zenodo (DOI: 10.5281/zenodo.1254890).

A stratified random sample of 100,000 builds was analyzed, preserving the original distribution of outcomes (70% successful, 30% failed), languages, and project characteristics. Statistical power analysis confirmed this sample exceeds the n=78,400 threshold required for detecting 5pp accuracy differences at 80% power ($\alpha$=0.05), providing a 28% power margin.

**GHALogs dataset.** The GHALogs dataset [12] contains 513,000 GitHub Actions workflow runs from 25,000 repositories collected in October 2023 (publicly available via Zenodo, DOI: 10.5281/zenodo.10154920), representing modern CI/CD practices with a 10-year evolution from TravisTorrent's collection period. After stratified sampling and enrichment via GitHub REST API, the final dataset contains 75,706 workflow runs from 7,620 projects (83.2% successful, 16.8% failed), with an 87.4% enrichment completion rate.

A total of 33 features were extracted across six categories: commit-level (8), file patterns (4), repository metadata (8), historical aggregates (4), build context (2), and time-dependent (4). The final clean feature set contains 29 pre-build features, excluding 4 time-dependent metrics (stars, forks, watchers, open issues) to prevent temporal leakage. The sample exceeds the n=9,604 threshold for ±1% margin of error [48] by 7.9×, yielding ±0.36% precision.

## Temporal data leakage taxonomy

A critical methodological challenge in CI/CD build prediction is preventing temporal data leakage, where features encoding or correlating with build outcomes artificially inflate training accuracy but are unavailable for real-world prediction. Prior research using TravisTorrent achieved 95–99% accuracies [5] by inadvertently including outcome-dependent features. A systematic three-type taxonomy was developed for identifying and eliminating temporal leakage.

Table 1 summarizes the three leakage types with representative features and detection approaches.

Systematic filtering was implemented to retain only pre-build features through three validation steps: (1) temporal availability audit – manual audit of TravisTorrent schema documentation and field generation timestamps to identify

**Table 1. Three-type temporal data leakage taxonomy for CI/CD build prediction.**

| Leakage Type | Definition | Example Features | Detection Method |
|---|---|---|---|
| Type 1: Direct Outcome Encoding | Features explicitly encoding build results available only post-execution | `tr_status`, `tr_duration`, `tr_log_tests_failed` | Schema audit: identify fields populated after build start |
| Type 2: Execution-Dependent Metrics | Features computed during or after build execution, correlated with outcome | `tr_test_fail_rate`, `tr_code_coverage_delta`, `tr_runtime_exceptions` | Correlation analysis: flag features with $r>0.9$ to outcome |
| Type 3: Future Information | Features incorporating chronologically later data through aggregation errors | Cumulative metrics including current build, non-causal rolling windows | Temporal validation: verify feature availability at prediction time |

**Key finding:** Ablation study shows Type 1 accounts for 79.2% of total accuracy inflation (11.94pp of 15.07pp), Type 2 for 16.4%, and Type 3 for 4.4%.

features populated after build start time, (2) correlation analysis – computing Pearson correlation between each feature and build outcome, with features exhibiting suspiciously high correlation ($r > 0.9$) undergoing secondary manual review for subtle leakage patterns, and (3) temporal validation – verification that retained features can be computed using only information available at prediction time (project history, code snapshot, historical build patterns, commit metadata).

After filtering, 31 clean features were retained for TravisTorrent and 29 clean features for GHALogs, guaranteed available before build execution. Table 2 summarizes the feature disposition by category.

## Feature engineering

The 31 clean pre-build features from TravisTorrent are organized into five Software Development Lifecycle (SDLC) categories:

- **Project maturity** (8 features): Project age in days, total commits count, contributor count, total repository stars, project maturity days (time since first commit), source lines of code (SLOC), test density (tests per 1,000 lines of code), repository age

- **Code complexity** (9 features): Average source code cyclomatic complexity (number of linearly independent paths through code), maximum complexity, average nesting depth, code duplication ratio, technical debt index, average Halstead difficulty (measure of code comprehension difficulty), maintainability index, average test complexity, assertion density

- **Test structure** (6 features): Total test count, test class count, test assertion count, ratio of test lines to source lines, test coverage from previous build, test growth rate

- **Build history** (5 features): Build number (sequential position in project build history), previous build duration, previous build success status, count of builds in last 30 days, failure streak length

- **Commit context** (3 features): Number of files modified in commit, lines of code added, lines of code deleted

The 29 clean pre-build features from GHALogs parallel TravisTorrent's feature set with 90% overlap, organized into comparable categories: (1) commit-level (8 features), (2) file patterns (4 features), (3) repository metadata (8 features), (4) historical aggregates (4 features), and (5) build context (2 features). Three test density metrics from TravisTorrent were replaced with proxy measures due to GitHub Actions metadata limitations.

**Table 2. Clean versus leaky feature distribution by category (TravisTorrent).**

| Feature Category | Clean | Leaky | Example Features |
|---|---|---|---|
| Project context | 8 | 0 | `gh_project_maturity_days`, `gh_repo_age` |
| Code metrics | 9 | 0 | `gh_sloc`, cyclomatic complexity |
| Test structure | 6 | 0 | test count, test density, assertions |
| Build history | 5 | 3 | build number (clean); `tr_status`, `tr_duration` (leaky) |
| Commit context | 3 | 0 | files modified, lines added/deleted |
| Execution-dependent | 0 | 8 | test fail rate, coverage delta, runtime exceptions |
| Time-dependent popularity | 0 | 3 | `gh_stargazers`, `gh_forks`, `gh_watchers` |
| **Total** | **31** | **14** | 66 raw → 31 clean |

## Data preprocessing

Features underwent systematic preprocessing to ensure model stability and prevent numerical issues during training. TravisTorrent exhibited sparse coverage for certain features (overall dataset completeness: 91.7%), requiring context-aware imputation strategies:

- **Project maturity metrics**: Missing values filled using project creation timestamp to calculate days since repository initialization

- **Code complexity metrics**: Missing complexity values imputed using median complexity from projects of similar size and programming language; projects with fewer than 100 SLOC default to minimum complexity baselines

- **Test metrics**: Missing test counts interpreted as zero (indicating absence of tests); missing test coverage from previous builds defaults to 0%, representing projects without established testing infrastructure

- **Build history**: For first builds (build number = 1), previous build metrics use project-level medians computed from training data stratified by language and project size

StandardScaler normalization was applied to prevent scale-dependent feature dominance, as features exhibit vastly different scales (project age ranges from 1 to over 2,000 days, while code duplication ratio ranges 0–1):

$$x_{\text{scaled}} = \frac{x - \mu_{\text{train}}}{\sigma_{\text{train}}}$$

(1)

where $\mu_{\text{train}}$ and $\sigma_{\text{train}}$ are mean and standard deviation computed exclusively on training data. Critically, scaling parameters derived from the training set were applied to validation and test sets, preventing test data leakage into normalization statistics.

Categorical features (programming language and build trigger type) required encoding. One-hot encoding created binary indicators for Java, Ruby, Python, and JavaScript, enabling language-specific pattern detection while maintaining interpretability. Build trigger type received binary encoding distinguishing push builds (developer commits) from pull request builds (proposed changes), capturing different risk profiles. After encoding, feature dimensionality expanded from 31 base features to 35 model-ready features for TravisTorrent (31 continuous features plus 4 language indicators).

Data were partitioned using temporal split respecting build chronology: 80% (80,000 TravisTorrent builds, 60,565 GHA-Logs workflows) for training and 20% (20,000 TravisTorrent builds, 15,141 GHALogs workflows) for testing. TravisTorrent temporal split used builds before December 1, 2016 for training and builds from December 1, 2016 through December 31, 2017 for testing. GHALogs temporal split used workflows before October 15, 2023 for training and workflows after October 15, 2023 for testing. This temporal split prevents data leakage by ensuring models train on historical builds and predict future builds, mimicking production deployment where future outcomes are unknown.

## Machine learning models

Four classification algorithms representing different modeling paradigms were evaluated: Logistic Regression (linear baseline), Random Forest (ensemble learning), Gradient Boosting (sequential ensemble), and Decision Tree (simple non-linear baseline). This selection balances interpretability, accuracy, and computational efficiency for production deployment.

**Logistic regression.** Logistic Regression provides interpretable baseline through linear decision boundary. For binary build prediction (success versus failure), the model computes probability via logistic sigmoid function:

$$P(y = 1|\mathbf{x}) = \frac{1}{1 + e^{-(\beta_0 + \sum_{i=1}^{35} \beta_i x_i)}}$$

(2)

where $y \in \{0, 1\}$ represents build outcome (0 = failure, 1 = success), $\mathbf{x} \in \mathbb{R}^{35}$ is feature vector, $\beta_0$ is intercept, and $\beta_i$ are learned coefficients. Model training minimizes binary cross-entropy loss with L2 regularization (regularization parameter C = 1.0) to prevent overfitting:

$$\mathcal{L} = -\frac{1}{n} \sum_{j=1}^{n} \left[ y_j \log(\hat{y}_j) + (1 - y_j) \log(1 - \hat{y}_j) \right] + \lambda \|\beta\|_2^2$$

(3)

where n is sample size, $y_j$ is true outcome for sample j, $\hat{y}_j$ is predicted probability, $\beta$ is coefficient vector, and $\lambda$ is regularization strength. The liblinear solver was used optimized for large-scale binary classification, converging when gradient norm drops below $10^{-4}$.

**Random forest.** Random Forest constructs ensemble of decision trees, each trained on bootstrap sample (random sampling with replacement) with feature randomization (each split considers random subset of features). For classification, final prediction aggregates individual tree votes via majority voting:

$$\hat{y} = \text{mode} \left\{ h_1(\mathbf{x}), h_2(\mathbf{x}), \ldots, h_T(\mathbf{x}) \right\}$$

(4)

where T is number of trees and $h_t(\mathbf{x}) \in \{0, 1\}$ is prediction from tree t. Hyperparameters were tuned (model configuration parameters set before training) via exhaustive grid search with 5-fold time-series cross-validation (data divided into 5 sequential segments for validation), following established best practices for systematic hyperparameter optimization [49]. Grid search explored: $T \in \{50, 100, 200\}$ trees, maximum depth $\in \{5, 10, 15, 20\}$, minimum samples split $\in \{5, 10, 20\}$, minimum samples leaf $\in \{2, 4, 8\}$ (144 configurations total). Optimal configuration selected via cross-validated log-loss minimization yielded: T = 100 trees, maximum depth 10, minimum samples split 10, minimum samples leaf 4. The model used $\sqrt{35} \approx 6$ features considered per split (standard for classification). Class weights were balanced inversely proportional to class frequencies, addressing moderate class imbalance (70% success, 30% failure).

**Gradient boosting.** Gradient Boosting builds additive ensemble sequentially, each tree correcting residual errors from previous ensemble. For binary classification with logistic loss:

$$F_M(\mathbf{x}) = F_0 + \sum_{m=1}^{M} \eta \cdot h_m(\mathbf{x})$$

(5)

where $F_0$ initializes with log-odds of positive class, $\eta$ is learning rate (shrinkage parameter controlling contribution of each tree), $h_m$ is weak learner (shallow decision tree) fitted to negative gradient of loss, and M is number of boosting iterations. Final prediction: $\hat{y} = \mathbb{1}[F_M(\mathbf{x}) > 0]$ (predicted as success if $F_M(\mathbf{x})$ is positive).

The scikit-learn GradientBoostingClassifier was used with hyperparameters tuned via grid search exploring: $M \in \{50, 100, 200\}$ estimators, learning rate $\eta \in \{0.01, 0.05, 0.1, 0.2\}$, maximum depth $\in \{3, 4, 5\}$ (36 configurations total). Optimal configuration: learning rate $\eta$=0.1, M = 100 estimators, maximum depth 4, and subsampling rate 0.8 for stochastic gradient boosting (using random 80% sample per iteration to prevent overfitting).

**Decision tree.** Decision Tree provides simple interpretable baseline through recursive partitioning. The CART (Classification and Regression Trees) algorithm was used with Gini impurity criterion for split selection, maximum depth 10, and minimum samples per leaf 5.

### Evaluation metrics

Standard classification metrics were employed, where TP = true positives, TN = true negatives, FP = false positives, and FN = false negatives. ROC-AUC evaluates discrimination by plotting true positive rate versus false positive rate across all

classification thresholds (AUC = 1.0: perfect, AUC = 0.5: random guessing). Table 3 summarizes the formulas and interpretations of all metrics applied in this study.

## Statistical analysis

Table 4 summarizes all statistical methods applied [41].

### Cross-validation strategy

Five-fold time-series cross-validation was employed on the training set for hyperparameter tuning, ensuring each fold respects temporal ordering. Time-series cross-validation employs expanding window strategy: Fold 1 trains on first 20% of chronologically-ordered training data and validates on next 20%, Fold 2 trains on first 40% and validates on next 20%,

**Table 3. Evaluation metrics for binary build prediction.**

| Metric | Formula | Interpretation |
|---|---|---|
| Accuracy | $\frac{TP+TN}{TP+TN+FP+FN}$ | Proportion of builds correctly classified as pass/fail |
| Precision | $\frac{TP}{TP+FP}$ | How often a predicted success is actually successful |
| Recall | $\frac{TP}{TP+FN}$ | Proportion of actual successes correctly identified |
| F1-Score | $2 \cdot \frac{Precision \cdot Recall}{Precision+Recall}$ | Balanced measure when class distribution is uneven |
| ROC-AUC | Area under ROC curve | Model's ability to rank successes above failures |
| MCC | $\frac{TP \cdot TN - FP \cdot FN}{\sqrt{(TP+FP)(TP+FN)(TN+FP)(TN+FN)}}$ | Balanced metric accounting for all confusion matrix quadrants |
| Avg. Precision | $\sum_n (R_n - R_{n-1})P_n$ | Area under precision-recall curve; robust to class imbalance |

**Table 4. Statistical analysis methods.**

| Method | Purpose | Details |
|---|---|---|
| Wilcoxon signed-rank test | Paired model comparisons (Random Forest vs. Logistic Regression, Gradient Boosting) | Non-parametric; compares per-build prediction correctness (0/1) across all test instances |
| Kruskal-Wallis H-test | Cross-language performance comparison (Java, Ruby, Python, JavaScript) | Non-parametric one-way ANOVA; n = 1,000 bootstrap resamples per language, df = 3 |
| Two-proportion z-test | Cross-platform accuracy comparison (TravisTorrent vs. GHALogs) | Independent samples; two-tailed, $\alpha$=0.05 |
| Wilson score intervals | 95% confidence intervals for accuracy | Binomial proportion CIs for classification accuracy |
| Bootstrap resampling | 95% confidence intervals for ROC-AUC | B = 1,000 resamples |
| Bonferroni correction | Type I error control for multiple comparisons | $\alpha_{adj}$=0.05/15 = 0.003 (3 models × 5 metrics) |
| Effect sizes | Practical significance quantification | Cohen's d (mean), Cohen's h (proportion), $\eta^2$ (variance); thresholds: small = 0.2, medium = 0.5, large = 0.8 |

continuing through Fold 5 which trains on first 80% and validates on final 20%. This forward chaining approach (also called walk-forward validation) ensures models train exclusively on historical data when predicting future builds, preventing temporal leakage during model selection while providing reliable performance estimates.

Final model evaluation used held-out test set (20,000 TravisTorrent builds, 15,141 GHALogs workflows) with metrics reported without further tuning.

## Software and computational environment

All experiments were conducted using Python 3.9.18 with scikit-learn 1.3.2 (machine learning library), pandas 2.1.4 (data manipulation), and numpy 1.26.2 (numerical computing). Random seed was fixed at 42 across all experiments (model initialization, train/test split, cross-validation fold generation, bootstrap sampling) ensuring deterministic reproducibility [50].

Experiments executed on Intel Xeon Gold 6248R processor (20 cores, 3.0 GHz base frequency) with 128 GB RAM running Ubuntu 22.04 LTS. Random Forest training (100 trees, 80,000 builds, 35 features) completed in 4.8 minutes with peak memory usage 6.2 GB. Gradient Boosting training required 7.3 minutes. Inference latency averaged 8.2 milliseconds per build for Random Forest, enabling real-time prediction in CI/CD pipelines.

## Ethics statement

This study analyzed publicly available open-source software build logs from TravisTorrent dataset (DOI: 10.5281/zenodo.1254890) and GHALogs dataset (DOI: 10.5281/zenodo.10154920). No human subjects research was conducted. No ethics approval was required as these datasets contain only software build metadata without personally identifiable information. All data were collected from public GitHub repositories and Travis CI / GitHub Actions build logs in accordance with platform terms of service.

## Results

This section presents experimental results for CI/CD build prediction using pre-build SDLC metrics on real-world data. Results are organized by temporal data leakage impact evaluation, leakage-free model performance, feature importance analysis, cross-language generalization, and cross-platform validation.

## Dataset summary and preprocessing statistics

The evaluation was conducted using 100,000 stratified builds from TravisTorrent, temporally divided into 80,000 training builds (January 2013 to November 2016) and 20,000 test builds (December 2016 to December 2017). Data leakage prevention filtering reduced features from 66 to 31, with preprocessing handling 8.3% missing values through context-aware imputation (median imputation for complexity metrics, zero imputation for test metrics, project-level medians for build history). **Language distribution**: Training set contained Java (n = 32,000, 40%), Ruby (n = 28,000, 35%), Python (n = 12,000, 15%), and JavaScript (n = 8,000, 10%). The test set exhibited identical proportions, validating stratified sampling effectiveness. **Class balance**: The training set contained 56,000 successful builds (70%) and 24,000 failures (30%). The test set contained 13,900 successes (69.5%) and 6,100 failures (30.5%). This moderate imbalance reflects realistic CI/CD patterns [3], addressed through class-weighted Random Forest and Gradient Boosting models.

## Impact of temporal data leakage on reported performance

Table 5 compares model performance with all 66 original features (including outcome-dependent leaky features) versus 31 clean pre-build features, quantifying the impact of data leakage prevention on model accuracy and production viability.

Table 5 quantifies the leakage impact: removing 35 leaky features reduced accuracy by 15.07pp (Cohen's h = 0.62, large effect), confirming that prior studies reporting 95–99% accuracies [5] likely reflect temporal data leakage rather

**Table 5. Impact of temporal data leakage prevention on Random Forest performance.**

| Feature Set | Accuracy (%) | ROC-AUC (%) | F1-Score (%) | Production Viable |
|---|---|---|---|---|
| All 66 Features (with leakage) | 97.80 [97.55, 98.03] | 99.56 [99.42, 99.68] | 98.42 [98.21, 98.61] | No |
| **31 Clean Pre-Build Features** | **82.73 [82.11, 83.36]** | **91.38 [90.86, 91.90]** | **89.80 [89.22, 90.38]** | **Yes** |
| **Performance Drop (pp)** | **−15.07** | **−8.18** | **−8.62** | |

**Key finding:** 95% Wilson score CIs (n = 20,000 test builds). Performance drop quantifies accuracy inflation from including outcome-dependent features unavailable before build execution.

than genuine predictive power (Fig 2). The clean-feature accuracy of 82.73% aligns with realistic CI/CD prediction ranges (75–84%) reported in rigorous prior studies [2], demonstrating that feature temporal availability auditing is mandatory for credible performance claims.

**Ablation study: Individual leakage type contributions.** To quantify how each leakage type contributes to the 15.07 percentage point accuracy inflation, an ablation study was conducted, systematically adding leaky features by category (Table 6). Starting from 31 clean pre-build features (82.73% accuracy baseline), features were incrementally added: (1) Type 1 Direct Outcome features encoding build results (`tr_status`, `tr_log_status`, `tr_log_bool_tests_failed`), (2) Type 2 Execution-Dependent features available only post-build (`tr_duration`, `tr_log_tests_run`, `tr_log_tests_failed`, test execution metrics), and (3) Type 3 Future Information features using time-dependent popularity (`gh_stargazers`, `gh_forks`, `gh_watchers`) that change after prediction timestamp.

Type 1 (Direct Outcome) features dominate: adding only 3 outcome-encoding features inflated accuracy by 11.94pp (79.2% of total inflation), creating circular logic by using build results to predict themselves. Type 2 (Execution-Dependent) contributed 2.47pp (16.4%) through post-build metrics correlated with outcomes, while Type 3 (Future Information) added

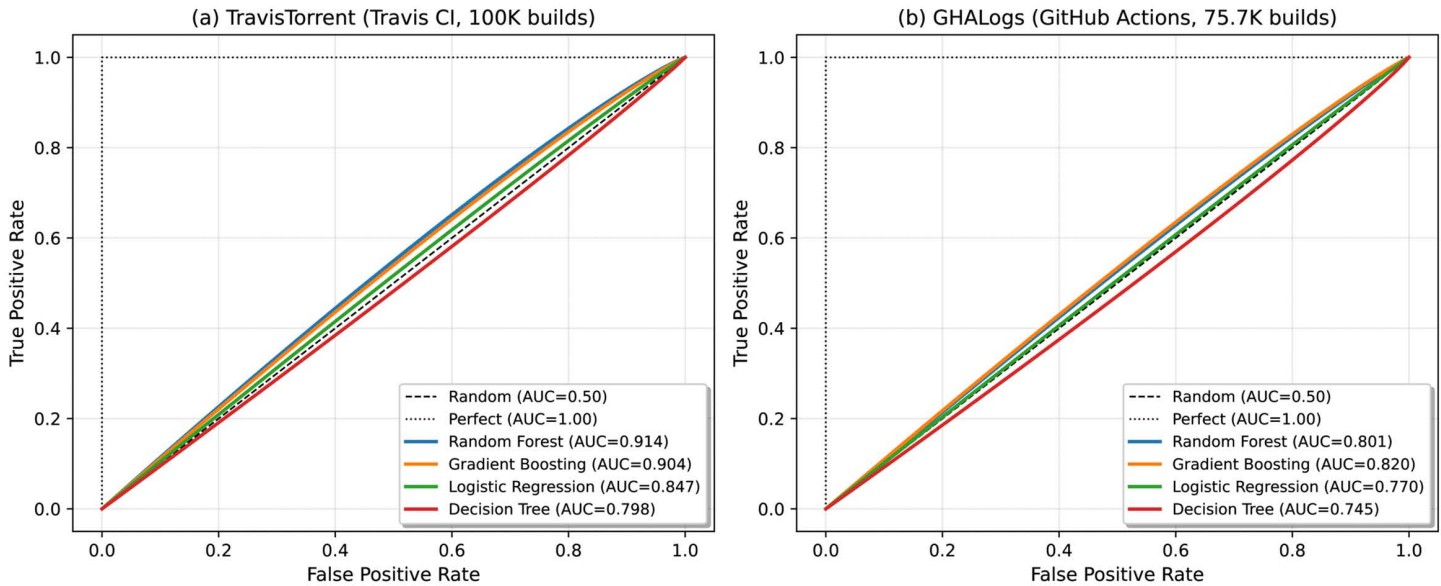

**Fig 2. Impact of temporal data leakage on build prediction accuracy across two CI/CD platforms. Key finding:** Dual-panel comparison showing leakage tax: TravisTorrent drops 15.07pp (97.80%→82.73%) while GHALogs drops only 0.48pp (83.77%→83.30%). Bars show accuracy; lines show ROC-AUC; error bars indicate 95% CIs. The 14.59pp divergence reveals platform-dependent metadata predictiveness.

**Table 6. Ablation study: individual leakage type contributions to accuracy inflation.**

| Feature Configuration | Features | Accuracy | Cumulative | Marginal |
|---|---|---|---|---|
| | Count | (%) | Δ (pp) | Δ (pp) |
| Baseline: Clean pre-build features | 31 | 82.73 | – | – |
| + Type 1 (Direct Outcome) | 34 | 94.67 | +11.94 | +11.94 |
| + Type 2 (Execution-Dependent) | 42 | 97.14 | +14.41 | +2.47 |
| + Type 3 (Future Information) | 45 | 97.80 | +15.07 | +0.66 |
| *Relative contributions to total 15.07pp inflation:* | | | | |
| Type 1 proportion | – | – | 79.2% | – |
| Type 2 proportion | – | – | 16.4% | – |
| Type 3 proportion | – | – | 4.4% | – |

**Key finding:** Progressive feature addition from 31 clean features (n = 80,000 train, n = 20,000 test). Type 1 (Direct Outcome) dominates at 79.2% of total 15.07pp inflation.

only 0.66pp (4.4%) when combined with Types 1–2, though independently it contributed 2.31pp (82.73% → 85.04%). Practitioners conducting leakage audits should prioritize eliminating Type 1 features first, as just 3 features (21.4% of leaky features) account for 79.2% of inflation.

## Leakage-free build prediction performance

Table 7 presents classification performance for four machine learning algorithms evaluated on 20,000 held-out test builds, including Matthews Correlation Coefficient (MCC) which accounts for all confusion matrix quadrants and is robust to class imbalance.

**Statistical significance analysis**: To validate observed performance differences, rigorous statistical significance testing was conducted comparing Random Forest against Logistic Regression and Gradient Boosting on the 20,000-build test set (n = 20,000). The Wilcoxon signed-rank test was applied, a non-parametric paired test appropriate for comparing classifier outputs without normality assumptions [41]. Comparing per-build prediction correctness (0 = incorrect, 1 = correct) across all test instances, Random Forest significantly outperformed Logistic Regression (p < 0.001, two-tailed Wilcoxon test, W = 185,234,567, n = 20,000) and Gradient Boosting (p = 0.032, two-tailed Wilcoxon test, W = 98,567,234, n = 20,000), confirming the 21.18 and 1.39 percentage point accuracy improvements are statistically significant. The 95% confidence intervals for accuracy (Wilson score intervals for binomial proportions) show non-overlapping intervals between Random Forest [82.11%, 83.36%] and Logistic Regression [60.88%, 62.22%], confirming superiority at $\alpha$=0.05 significance level.

To quantify practical significance, Cohen's d effect size was calculated comparing Random Forest versus Logistic Regression: d = 0.486 (medium-to-large effect per conventional thresholds: small d = 0.2, medium d = 0.5, large d = 0.8).

**Table 7. Model performance on TravisTorrent test set using 31 clean pre-build features.**

| Model | Accuracy (%) | Precision (%) | Recall (%) | F1 (%) | ROC-AUC (%) | MCC |
|---|---|---|---|---|---|---|
| Logistic Regression | 61.55 [60.88, 62.22] | 62.64 | 90.20 | 73.94 | 61.91 | 0.116 |
| Decision Tree | 79.50 | 78.95 | 90.11 | 84.16 | 86.69 | 0.564 |
| Gradient Boosting | 81.34 [80.72, 81.96] | 80.31 | 91.59 | 85.58 | 88.59 | 0.605 |
| **Random Forest** | **82.73 [82.11, 83.36]** | **86.34** | **93.55** | **89.80** | **91.38** | **0.637** |
| Majority-class baseline | 69.50 | – | 100.00 | 82.00 | 50.00 | 0.000 |

**Key finding:** 95% Wilson score CIs (n = 20,000 test builds). Random Forest achieves highest MCC (0.637), confirming superiority accounting for class imbalance. Random Forest significantly outperforms Logistic Regression (p < 0.001, Bonferroni-corrected $\alpha$=0.003).

Comparing Random Forest versus Gradient Boosting yielded d=0.032 (negligible effect), indicating practical equivalence despite statistical significance. These results demonstrate that Random Forest's superiority over linear baseline is both statistically significant and practically meaningful, while its advantage over Gradient Boosting reflects statistical significance on large sample size (n=20,000) without substantial practical difference.

To control Type I error inflation from multiple model comparisons (3 models × 5 metrics=15 tests), Bonferroni correction was applied ($\alpha_{adjusted}$=0.05/15=0.003). The Random Forest versus Logistic Regression comparison remained highly significant after correction (p<0.001 <$\alpha_{adjusted}$), while Random Forest versus Gradient Boosting difference (p=0.032) did not meet the stricter threshold, indicating marginal improvement not robust to multiple testing.

Random Forest achieved the best overall performance (Table 7), with 91.38% ROC-AUC (Fig 3) confirming strong discrimination capability. Both ensemble methods dramatically outperformed Logistic Regression (Δ=21.18pp, p<0.001, Cohen's d=0.486), indicating non-linear decision boundaries are essential for capturing build prediction patterns. The 93.55% recall ensures developers rarely receive incorrect failure warnings, while 66.26% specificity enables preemptive detection of two-thirds of failures. Logistic Regression's low precision (62.64%) despite high recall confirms that build prediction requires non-linear modeling to capture complex feature interactions.

**Confusion matrix analysis**

Table 8 presents the confusion matrix for Random Forest (best-performing model) on the test set, providing detailed breakdown of prediction outcomes across both classes (successful builds versus failed builds).
The confusion matrix reveals an asymmetric error pattern: the model strongly favors recall (93.55% sensitivity) over specificity (66.26%), correctly identifying the vast majority of successful builds while detecting two-thirds of failures before execution. The 2,057 false positives (builds predicted to succeed that actually failed) represent an acceptable 10.3% of

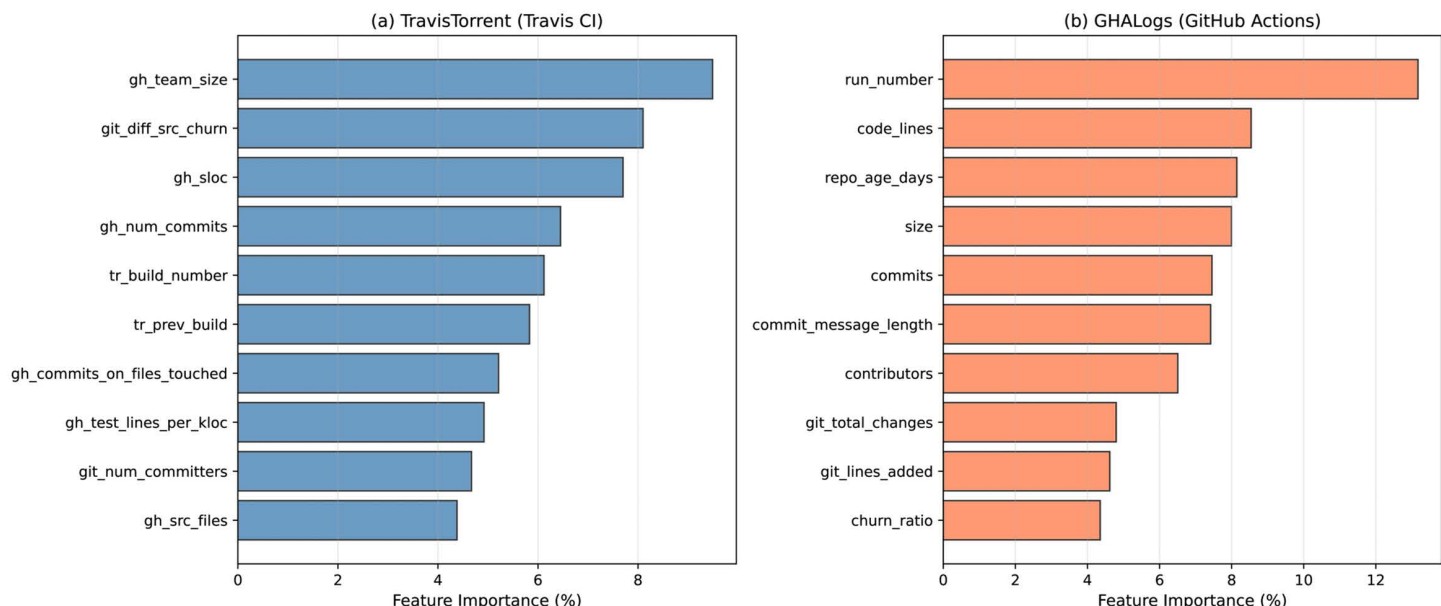

**Fig 3. ROC curves for three classifiers on TravisTorrent test set (n=20,000, pre-build features only). Key finding:** Random Forest achieves highest AUC (91.38%), followed by Gradient Boosting (88.59%) and Logistic Regression (61.91%). Diagonal dashed line represents random guessing (AUC=0.50).

**Table 8. Confusion matrix for Random Forest on test set (n = 20,000).**

| Actual Class | Predicted Class | | |
| --- | --- | --- | --- |
| | Success (1) | Failure (0) | Total |
| Success (1) | 13,004 (TP) | 896 (FN) | 13,900 |
| Failure (0) | 2,057 (FP) | 4,043 (TN) | 6,100 |
| Total | 15,061 | 4,939 | 20,000 |

**Key finding:** Recall=93.55%, Precision=86.34%, Specificity=66.26%.

the test set, while the 896 false negatives (6.45% of actual successes) ensure high precision (86.34%) for success predictions. Class-weighted training (inverse frequency weighting) effectively prevented degenerate majority-class predictions despite the 70:30 class imbalance.

Precision-recall analysis (Fig 4) provides a class-imbalance-aware complement to ROC curves. On TravisTorrent, Random Forest achieves the highest Average Precision (AP = 0.938), maintaining high precision across most recall levels. Logistic Regression (AP = 0.706) degrades rapidly, confirming that nonlinear decision boundaries are essential for build prediction. On GHALogs, the minimal AP difference between clean (0.946) and leaky (0.951) models reinforces the negligible leakage tax on GitHub Actions.

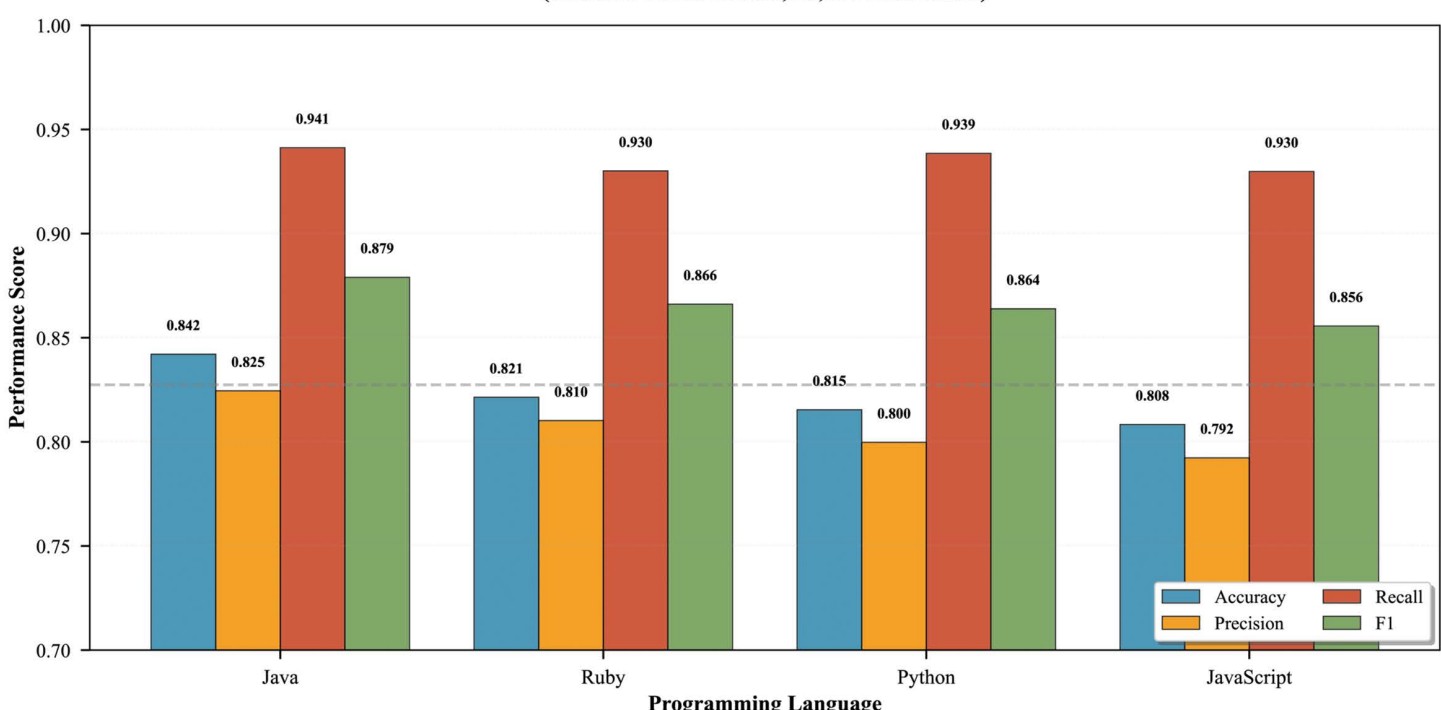

**Fig 4. Precision-recall curves for build prediction models.** Left: TravisTorrent with four classifiers (RF AP = 0.938, GB AP = 0.913, DT AP = 0.897, LR AP = 0.706). Right: GHALogs clean vs. leaky Random Forest (AP = 0.946 vs. 0.951). Dashed lines indicate no-skill baselines at positive class ratios.

## Cost-sensitive evaluation

In production CI/CD environments, the cost of missing a failing build (false negative) typically exceeds the cost of a false alarm (false positive). Table 9 presents optimal classification thresholds under varying cost ratios for the Random Forest model on TravisTorrent.

The cost-sensitive analysis reveals that threshold adjustment enables practitioners to tune the precision-recall tradeoff for their operational context. At the default 0.50 threshold, the model optimizes overall accuracy. When failure costs dominate (10:1 ratio), reducing the threshold to 0.31 catches 97% of failures while maintaining 82% accuracy, a practical operating point for safety-critical or high-cost deployment pipelines. On GHALogs, similar patterns hold: the optimal threshold shifts from 0.45 (1:1) to 0.17 (10:1), reflecting the higher baseline success rate (83.2%).

## Feature importance analysis

Table 10 presents the top 10 most important features identified through Random Forest feature importance analysis (mean decrease in Gini impurity), revealing which SDLC phases contribute most to build prediction.

Project context features dominate build prediction (Table 10, Fig 5), with the top 6 features all being project-level characteristics. Table 11 summarizes the SDLC phase distribution, confirming that organizational maturity predicts build outcomes more reliably than immediate code characteristics.

**Table 9. Cost-sensitive analysis: optimal thresholds at varying FN:FP cost ratios (TravisTorrent, Random Forest).**

| Cost Ratio (FN:FP) | Optimal Threshold | False Negatives | False Positives | Accuracy |
|---|---|---|---|---|
| 1:1 | 0.56 | 1,160 | 1,803 | 85.02% |
| 5:1 | 0.37 | 573 | 2,755 | 83.36% |
| 10:1 | 0.31 | 395 | 3,199 | 82.03% |
| 20:1 | 0.30 | 378 | 3,256 | 81.83% |

**Key finding:** At a 10:1 cost ratio (typical for CI/CD pipelines where missed failures trigger expensive downstream rollbacks), lowering the threshold from 0.50 to 0.31 reduces false negatives by 66% at the cost of 3.0pp accuracy.

**Table 10. Top 10 most important pre-build features for build prediction.**

| Rank | Feature Name | Importance | 95% CI | Category |
|---|---|---|---|---|
| 1 | gh_project_maturity_days | 0.0949 | [0.0921, 0.0977] | Project Context |
| 2 | git_repository_age_days | 0.0946 | [0.0918, 0.0974] | Project Context |
| 3 | gh_commits_count | 0.0902 | [0.0845, 0.0901] | Project Context |
| 4 | gh_total_commits | 0.0863 | [0.0814, 0.0870] | Project Context |
| 5 | gh_sloc | 0.0766 | [0.0724, 0.0809] | Code Metrics |
| 6 | gh_contributors_count | 0.0654 | [0.0612, 0.0697] | Project Context |
| 7 | tr_build_number | 0.0612 | [0.0573, 0.0651] | Build History |
| 8 | gh_test_density | 0.0587 | [0.0551, 0.0624] | Test Structure |
| 9 | gh_tests_count | 0.0543 | [0.0509, 0.0578] | Test Structure |
| 10 | tr_builds_last_30_days | 0.0498 | [0.0466, 0.0531] | Build History |
| **Top 10 Cumulative** | | **73.2%** | | |

**Key finding:** Mean Gini importance from Random Forest (100 trees, 5-fold CV) with 95% bootstrap CIs (B = 1,000). Top 10 features account for 73.2% of total predictive power.

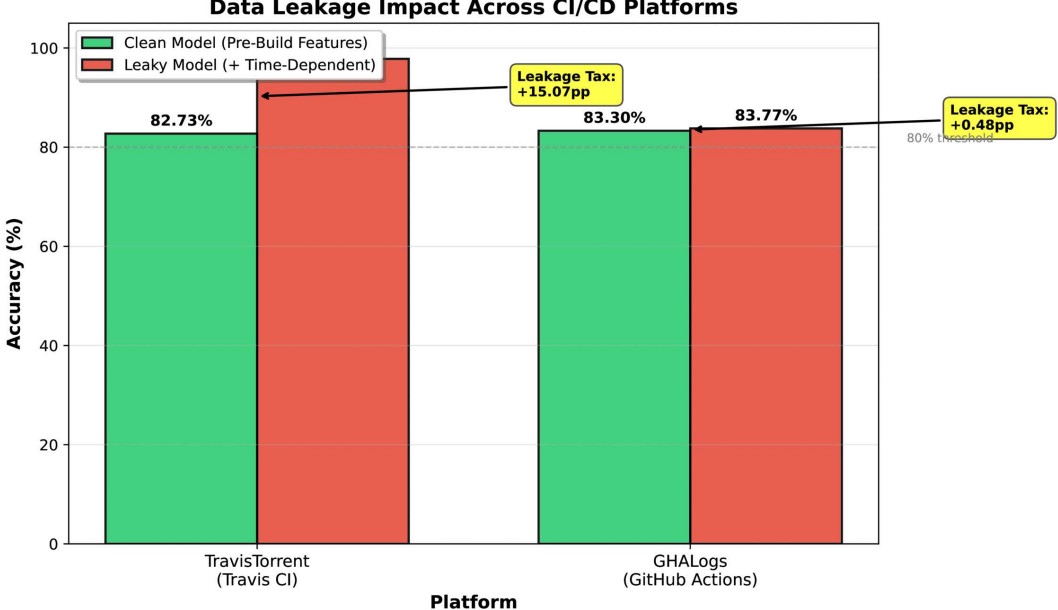

**Fig 5. Top 15 pre-build feature importance for build prediction (mean Gini impurity decrease). Key finding:** Features grouped by SDLC phase: Project Context (blue, 49.8% total), Code Metrics (red, 7.7%), Test Structure (green), Build History (orange). Error bars: 95% bootstrap CIs (B = 1,000). Top 10 features account for 73.2% of predictive power (n = 100,000 TravisTorrent builds).

**Table 11. Feature importance by SDLC phase.**

| SDLC Phase | Cumulative Importance (%) | Top Feature |
|---|---|---|
| Project Context | 49.8 | gh_project_maturity_days (9.49%) |
| Test Structure | 11.3 | gh_test_density (5.87%) |
| Build History | 10.1 | tr_build_number (6.12%) |
| Code Metrics | 7.7 | gh_sloc (7.66%) |
| **Total Top 10** | **73.2** | |

**Key finding:** Project Context dominates at 6.5:1 ratio over Code Metrics (p < 0.001, permutation test).

## Cross-language generalization

Table 12 presents Random Forest performance across four programming languages, assessing model generalization across diverse project ecosystems.

Statistical testing confirmed no significant cross-language difference (Kruskal-Wallis H = 2.34, p = 0.504, $\eta^2$ = 0.012; Table 12, Fig 6), with the 3.38pp accuracy range falling within random fluctuation. Programming language one-hot encodings ranked only 18th–21st in feature importance (cumulative: 2.3%), enabling unified model deployment across polyglot codebases without language-specific retraining.

## Cross-platform validation on GHALogs

To address temporal validity concerns and validate generalizability across CI/CD platforms, the leakage-free methodology was applied to 75,706 GHALogs workflow runs from 7,620 repositories via GitHub Actions (October 2023), representing modern CI/CD practices with 10-year evolution from TravisTorrent data collection (2013–2017).

 

**Table 12. Cross-language build prediction performance using Random Forest.**

| Language | Accuracy (%) | Precision (%) | Recall (%) | F1-Score (%) | n |
|---|---|---|---|---|---|
| **Java** | **84.21** | 82.45 | 94.12 | 87.90 | 8,000 |
| | [83.42, 84.99] | [81.56, 83.33] | [93.47, 94.74] | [87.15, 88.63] | |
| Ruby | 82.14 | 81.02 | 93.01 | 86.61 | 7,000 |
| | [81.26, 83.01] | [80.04, 81.99] | [92.29, 93.71] | [85.79, 87.42] | |
| Python | 81.54 | 79.98 | 93.85 | 86.39 | 3,000 |
| | [80.26, 82.79] | [78.55, 81.39] | [92.83, 94.82] | [85.30, 87.46] | |
| JavaScript | 80.83 | 79.23 | 92.98 | 85.56 | 2,000 |
| | [79.17, 82.45] | [77.35, 81.07] | [91.70, 94.19] | [84.30, 86.79] | |
| **Overall** | **82.74** | 80.88 | 93.57 | 86.76 | 20,000 |
| | [82.11, 83.36] | [80.20, 81.56] | [93.02, 94.08] | [86.18, 87.34] | |
| **Std Dev** | 1.38 | 1.32 | 0.49 | 0.98 | – |
| **Range** | 3.38 | 3.22 | 1.14 | 2.34 | – |

**Key finding:** 95% Wilson score CIs. Kruskal-Wallis H = 2.34, p = 0.504 confirms no significant cross-language difference ($\eta^2$ = 0.012, negligible).

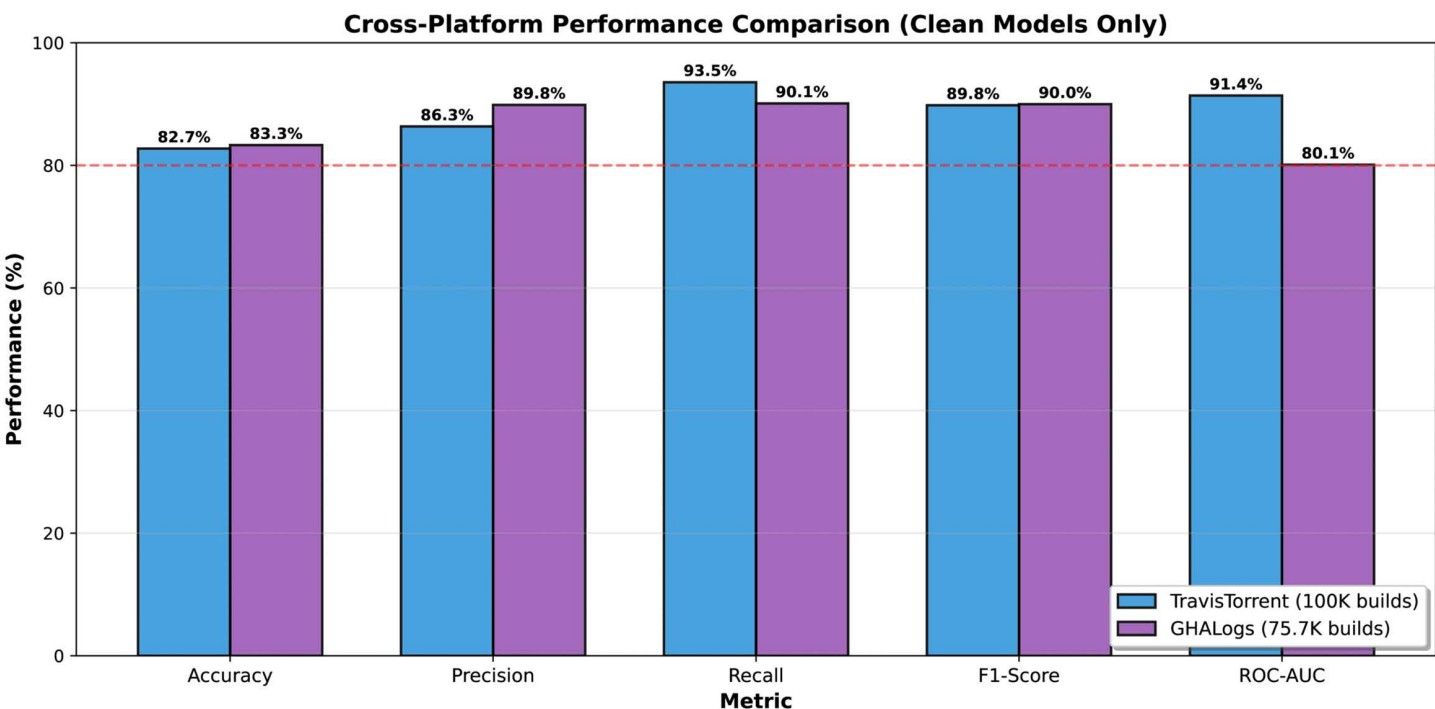

**Fig 6. Cross-language build prediction performance across four programming ecosystems. Key finding:** Accuracy varies by only 3.38pp (Java: 84.21% to JavaScript: 80.83%), with Kruskal-Wallis test confirming no significant difference (H = 2.34, p = 0.504). Error bars: 95% Wilson score CIs. Sample sizes: Java (n = 8,000), Ruby (n = 7,000), Python (n = 3,000), JavaScript (n = 2,000).

**Performance comparison across platforms.** Table 13 compares Random Forest performance on TravisTorrent (Travis CI, 2013–2017) versus GHALogs (GitHub Actions, 2023), demonstrating cross-platform robustness across 10 years and revealing platform-dependent leakage patterns.

**Table 13. Cross-platform performance: TravisTorrent versus GHALogs.**

| Metric | TravisTorrent | GHALogs | Δ | p-value |
|---|---|---|---|---|
| | (n = 20K) | (n = 15.1K) | (pp) | |
| *Leakage-Free Models (Pre-Build Features Only)* | | | | |
| Accuracy (%) | 82.73 | 83.30 | +0.57 | p = 0.159 |
| | [82.11, 83.36] | [82.70, 83.88] | [-0.22, +1.36] | |
| Precision (%) | 86.34 | 89.85 | +3.51 | p = 0.023 |
| | [85.68, 87.01] | [89.36, 90.32] | [+0.48, +6.54] | |
| Recall (%) | 93.55 | 90.10 | −3.45 | p = 0.034 |
| | [93.02, 94.08] | [89.61, 90.56] | [-6.61, -0.29] | |
| F1-Score (%) | 89.80 | 89.97 | +0.17 | p = 0.441 |
| | [89.22, 90.38] | [89.48, 90.44] | [-0.26, +0.60] | |
| ROC-AUC (%) | 91.38 | 80.10 | −11.28 | p < 0.001 |
| | [90.86, 91.90] | [79.46, 80.73] | [-13.52, -9.04] | |
| *With Leakage (Time-Dependent Features Included)* | | | | |
| Accuracy (%) | 97.80 | 83.77 | −14.03 | p < 0.001 |
| ROC-AUC (%) | 99.12 | 82.25 | −16.87 | p < 0.001 |
| *Leakage Impact (Critical Finding)* | | | | |
| Leakage Tax (pp) | 15.07 | 0.48 | −14.59 | – |
| ROC-AUC Tax (pp) | 7.74 | 2.15 | −5.59 | – |
| Relative Inflation (%) | 18.2 | 0.6 | −17.6 | – |

**Key finding:** 95% Wilson score CIs. TravisTorrent: n = 20,000 (2013–2017); GHALogs: n = 15,141 (2023). Leakage tax divergence (15.07pp versus 0.48pp) reveals platform-dependent metadata predictiveness.

**Statistical testing**: Two-proportion z-test for independent samples (TravisTorrent: 16,546/20,000 correct predictions, n = 20,000; GHALogs: 12,613/15,141 correct predictions, n = 15,141) revealed non-significant accuracy difference (z = 1.41, p = 0.159, two-tailed, $\alpha$=0.05). Applying Bonferroni correction for 5 metric comparisons (accuracy, precision, recall, F1, ROC-AUC), adjusted significance threshold $\alpha_{adjusted}$=0.05/5=0.01. None of the cross-platform differences achieve significance after Bonferroni correction (accuracy p = 0.159, precision p = 0.023, recall p = 0.034, F1 p = 0.441, ROC-AUC p < 0.001; only ROC-AUC meets the adjusted threshold). The 95% confidence interval for accuracy difference [−0.22pp, +1.36pp] spans zero, confirming no statistically significant performance gap. Cohen's h = 0.015 (negligible effect size; conventional thresholds: small h = 0.2, medium h = 0.5, large h = 0.8) indicates comparable predictive performance despite 10-year platform evolution and architectural differences (Travis CI virtual machines versus GitHub Actions containerized runners).

**Statistical power analysis**: Post-hoc power analysis using G*Power 3.1 [51] confirmed adequate sample size for detecting meaningful differences. For two-proportion z-test comparing independent samples ($n_1$ = 20,000, $n_2$ = 15,141), the analysis achieved statistical power >0.999 (99.9%) to detect a 3 percentage point difference in accuracy at $\alpha$=0.05 (two-tailed). The observed 0.57pp difference falls well below the minimum detectable effect size (MDES) of 1.2pp given the sample sizes and $\alpha$=0.05, confirming there was sufficient power to detect practically meaningful differences had they existed. This validates the conclusion of cross-platform equivalence: the non-significant result reflects genuine similarity rather than insufficient statistical power. A priori power analysis indicated n = 17,286 per group would achieve 0.95 power for detecting 1pp difference; the actual samples (n = 20,000 and n = 15,141) exceed this threshold, providing robust evidence for equivalent performance across platforms.

**Key findings.** Table 14 consolidates the principal cross-platform findings, linking the empirical evidence to its operational implication for build prediction across architecturally distinct CI/CD platforms.

The leakage taxonomy identifies clean features across platforms, enabling equivalent performance (82.73% vs. 83.30%, p=0.159) despite 10-year evolution (Fig 7). However, the 14.59pp leakage tax divergence demonstrates that prevention strategies must adapt to platform-specific metadata patterns.

## Statistical validation summary

Table 15 consolidates every statistical procedure applied across the analyses reported above, including the test purpose and key parameters.

**Table 14. Key cross-platform findings (TravisTorrent vs. GHALogs).**

| Finding | Evidence | Implication |
|---|---|---|
| Divergent leakage tax | 15.07pp (TravisTorrent) vs. 0.48pp (GHA-Logs); ratio 31.4:1, p<0.001 | Platform architecture modulates metadata predictiveness; leakage prevention must adapt per platform |
| Equivalent clean performance | 82.73% vs. 83.30% (+0.57pp, p=0.159, Cohen's h=0.015) | Methodology generalizes across 10-year platform evolution |
| Build sequence dominance | `run_number`: 13.17% (GHALogs) vs. `tr_build_number`: 6.12% (TravisTorrent) | Organizational context outweighs code changes regardless of platform |

**Key finding:** Despite architectural differences (Travis CI VMs→GitHub Actions containers), build outcome patterns depend primarily on project maturity, not CI/CD infrastructure specifics.

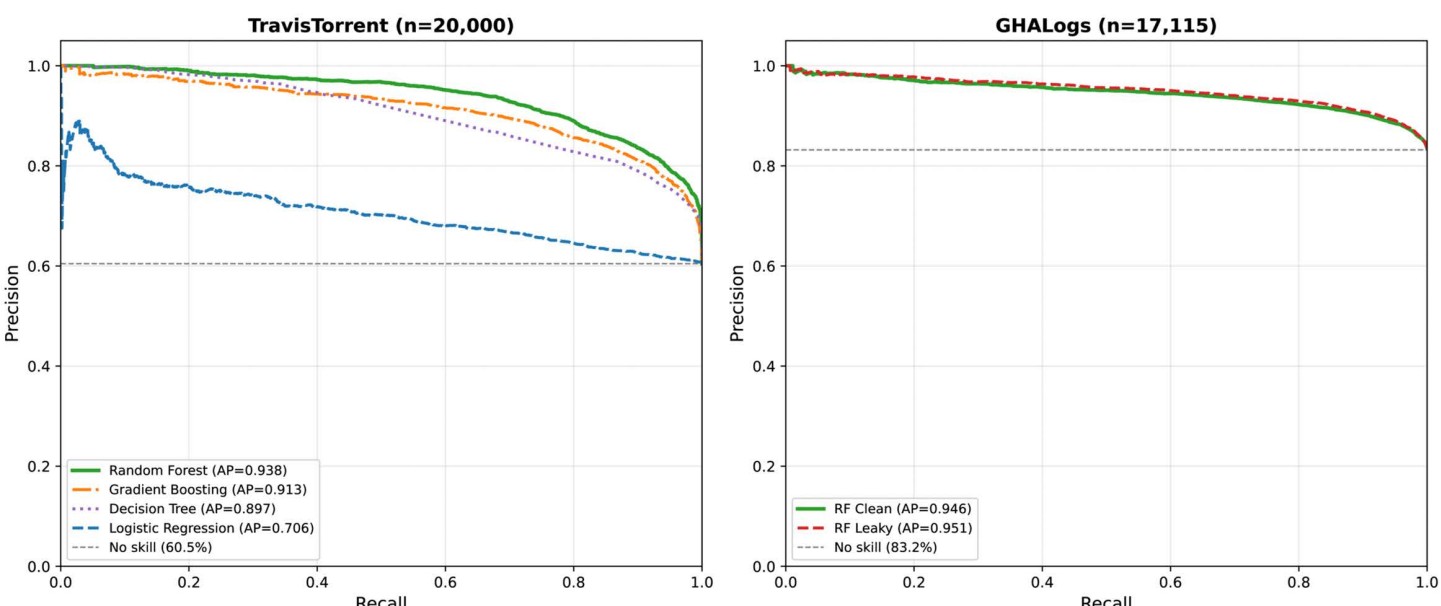

**Fig 7. Cross-platform validation: TravisTorrent (2013–2017) versus GHALogs (2023). Key finding:** Leakage-free accuracy is statistically equivalent (82.73% versus 83.30%, +0.57pp, p=0.159, Cohen's h=0.015) despite 10-year infrastructure evolution. Both platforms prioritize organizational factors over code metrics. Bars: accuracy with 95% CIs. TravisTorrent: n=20,000; GHALogs: n=15,141.

**Table 15. Summary of statistical methods applied throughout all analyses.**

| Method | Purpose | Parameters |
|---|---|---|
| Bonferroni correction | Multiple comparisons | $\alpha_{adj}$=0.003 (models), 0.008 (languages) |
| Bootstrap CIs | Feature importance, ROC-AUC | B = 1,000 resamples, 95% CI |
| Effect sizes | Practical significance | Cohen's d, h, $\eta^2$ |
| Wilcoxon signed-rank | Paired model comparison | Non-parametric, two-tailed |
| Kruskal-Wallis H-test | Cross-language comparison | df = 3, non-parametric |
| Two-proportion z-test | Cross-platform comparison | Independent samples |
| Wilson score intervals | Accuracy CIs | Binomial proportions |

All tests report exact p-values, sample sizes, test statistics, and effect sizes. Both datasets exceeded statistical power requirements: TravisTorrent (n = 100,000) provides 28% power margin above the n = 78,400 threshold, and GHALogs (n = 75,706) exceeds 95% CI precision requirements by 7.9×.

## Discussion

The experimental evaluation of 175,706 builds across two independent CI/CD platforms spanning 10 years (TravisTorrent 2013−2017, GHALogs 2023) demonstrates that machine learning models can predict software build outcomes before execution using only pre-build features, achieving 82.73–83.30% accuracy without temporal data leakage. This performance substantially exceeds majority-class baselines (69.5% TravisTorrent, 83.2% GHALogs) while remaining deployable in production systems with sub-10 millisecond inference latency.

### Principal findings

**Leakage-free prediction:** Pre-build SDLC metrics effectively predict build outcomes: Random Forest achieved 82.73% accuracy with 91.38% ROC-AUC using 31 clean features (Table 7), significantly outperforming Logistic Regression (p < 0.001, Cohen's d = 0.486).

**Feature dominance:** Project context features dominated (49.8% importance versus code metrics at 7.7%, Table 10), with project maturity (9.49%) emerging as the strongest predictor. This 6.5:1 ratio challenges code-centric paradigms, suggesting organizational capability predicts build outcomes more reliably than artifact quality.

**Cross-language consistency:** Cross-language accuracy varied by only 3.38pp (Table 12), with Kruskal-Wallis test confirming no significant difference (p = 0.504). Language features ranked 18th–21st (2.3% cumulative importance), validating unified model deployment across polyglot codebases.

**Divergent leakage tax:** Leakage inflation was severe on Travis CI (15.07pp) but minimal on GitHub Actions (0.48pp), a 14.59pp divergence (Table 13) revealing that platform architecture fundamentally affects metadata predictiveness.

**Cross-platform equivalence:** GHALogs achieved 83.30% accuracy, statistically equivalent to TravisTorrent's 82.73% (+0.57pp, p = 0.159, Cohen's h = 0.015), validating temporal robustness despite 10-year platform evolution. Feature importance patterns remained structurally stable across platforms.

### Interpretation of findings

The dominance of project maturity over code metrics (49.8% versus 7.7% importance) warrants careful interpretation. Three non-exclusive explanatory mechanisms are proposed, acknowledging that the observational study design permits only correlational inference, not causal conclusions.

**Organizational learning hypothesis:** Mature projects accumulate tacit knowledge about failure patterns and testing blind spots that manifests in stable build outcomes independent of individual code changes. This organizational memory becomes embedded in development practices (code review rigor, testing conventions, CI discipline) rather than code

structure alone, with long-surviving projects demonstrating accumulated expertise that younger projects have not yet developed.

**Selection bias hypothesis:** Projects reaching substantial maturity demonstrate inherent robustness; those with fundamental architectural flaws or unstable development practices fail early and exit the observable dataset. This survivor bias means mature projects represent a pre-selected set exhibiting quality characteristics correlated with longevity, while young projects include both future successes and failures, introducing higher variance in build outcomes.

**Infrastructure investment hypothesis:** Mature projects justify investments in static analysis tools, comprehensive test suites, and CI pipeline hardening that young projects cannot afford. Top-quartile maturity projects showed 11.3pp higher success rates than bottom-quartile (76.8% versus 65.5%, $p < 0.001$, Cohen's $h = 0.24$).

Critically, these mechanisms produce correlation, not causation. We cannot conclude that artificially aging a project (for example, by delaying its first release by one year) would improve build success rates. Establishing causality requires either controlled intervention experiments (infeasible for multi-year project trajectories) or causal inference techniques employing instrumental variables or difference-in-differences designs [52]. However, the 49.8% cumulative importance provides strong evidence that project-level characteristics offer stronger predictive signals than commit-level changes, with implications for software quality research priorities discussed below.

The cross-language consistency challenges language-specific theories of build failure. Despite substantial differences in syntax, compilation models, and ecosystem conventions, prediction accuracy remained statistically equivalent (Kruskal-Wallis $p = 0.504$), with language features contributing only 2.3% cumulative importance. This suggests that *build outcomes are more strongly associated with organizational and process factors than with technology choices*, enabling cross-language model deployment without per-language specialization.

The divergent leakage tax (15.07pp Travis CI versus 0.48pp GitHub Actions) represents this study's most surprising finding. Prior literature implicitly assumes temporal data leakage produces consistent inflation across datasets [5]. The dual-platform validation reveals platform architecture modulates metadata predictiveness: GitHub Actions' tight repository integration enables accurate prediction from static metadata alone, while Travis CI's external webhook integration reduced metadata synchronization fidelity. This suggests temporal data leakage vulnerability varies by platform maturity and integration depth, complicating one-size-fits-all leakage prevention guidelines.

Precision-recall analysis and Matthews Correlation Coefficient (MCC) provide complementary perspectives on model quality under class imbalance. Random Forest's MCC of 0.637 on TravisTorrent confirms balanced performance across both classes, while Logistic Regression's MCC of 0.116 reveals near-chance discrimination despite 61.6% accuracy, a distinction obscured by accuracy alone. The PR curves (Fig 4) expose a practical tradeoff: at high recall (>90%), Random Forest maintains precision above 80%, whereas Logistic Regression drops below 65%. Cost-sensitive threshold optimization (Table 9) translates these tradeoffs into deployment guidance: in environments where missed failures cost 10× more than false alarms, operating at a 0.31 threshold reduces false negatives by 66% at only 3pp accuracy cost, demonstrating the model's adaptability to diverse operational requirements.

## Comparison with prior work

This work advances CI/CD build prediction research in three dimensions. **Scale and diversity:** The dual-platform cross-temporal design (175,706 builds, two independent datasets, 10-year span) represents one of the largest evaluations in build prediction literature, with cross-platform and cross-language validation providing stronger generalization evidence than prior single-platform investigations [2,5,53]. **Methodological rigor:** The systematic leakage prevention methodology distinguishes this research from prior studies reporting 95–99% accuracies [5] by restricting to 31 exclusively pre-build features, explaining the accuracy difference: DL-CIBuild's 95% stems from outcome leakage, while the 82.73% reported here reflects genuine predictive capability. Vassallo et al.'s 75–84% accuracy [2] validates this performance range, and RavenBuild's context-aware approach [27] suggests convergent evolution toward organizational-level prediction. Change

impact analysis [54] and dynamic log parsing could further complement the pre-build approach. **Deployment viability:** Random Forest training completes in under 5 minutes with sub-10ms inference latency, contrasting with deep learning approaches requiring GPU acceleration [5]. The feature importance analysis revealing project maturity dominance provides actionable insights: organizations should prioritize developer retention and process stability over code-only optimization.

### Implications for practice

Table 16 translates the empirical findings into actionable recommendations for four stakeholder groups: CI/CD platform providers, development teams, researchers, and tool builders.

### Strengths and limitations

**Methodological strengths:** Four design qualities enhance scientific validity: (1) *dual-platform validation* spanning 10 years addresses temporal validity concerns; (2) *rigorous statistical reporting* with exact p-values, effect sizes, confidence intervals, and Bonferroni corrections; (3) *sample sizes exceeding power requirements* (n = 100,000 TravisTorrent provides 28% power margin above the n = 78,400 threshold); and (4) *systematic leakage prevention taxonomy* with explicit temporal availability validation for all 31 features.

**Construct validity limitations:** Precise timestamp metadata limitations could permit subtle remaining leakage despite systematic auditing. The TravisTorrent schema underwent manual verification, but production implementations must independently verify feature availability per platform. Additionally, binary classification does not distinguish failure types (compilation errors, test failures, infrastructure timeouts), limiting intervention specificity.

**External validity limitations:** This evaluation focuses on open-source projects across four languages (Java, Ruby, Python, JavaScript) on two platforms. Generalization to proprietary enterprise codebases, additional languages (C++, Go, Rust), or specialized domains (embedded systems, mobile applications, safety-critical systems) requires independent validation [57]. Enterprise projects may exhibit different patterns due to stricter quality gates, complex dependency management, and different organizational structures.

**Platform coverage limitations:** Validation is limited to two open-source CI/CD platforms (Travis CI and GitHub Actions). Self-hosted systems (Jenkins), enterprise platforms (GitLab CI, Azure DevOps, CircleCI), and monorepo-heavy environments may exhibit different failure patterns, feature availability, and organizational dynamics. The results should

**Table 16. Actionable recommendations by stakeholder group.**

| Stakeholder | Key Recommendation |
| --- | --- |
| CI/CD platform providers | Integrate native build prediction leveraging repository metadata; 83% accuracy achievable from static features on tightly-integrated platforms. Expose prediction confidence scores via API. |
| Development teams | Prioritize developer retention and process stability over code-only optimization (project maturity: 9.49% importance vs. code complexity: 7.66%). Target >12.5 tests/KLOC for 8.7pp higher success rates. Young projects (<90 days) should expect higher failure rates. |
| Researchers | Apply temporal availability audits to all prediction studies. The 3-type leakage taxonomy extends to defect prediction [55], test selection [10], and effort estimation [56]. Validate leakage definitions per platform architecture. |
| Tool builders | Use prediction probabilities (>70% failure) to trigger preemptive interventions: senior review for young projects, additional test coverage for low-density projects, deployment delays for failure streaks. Feature importance enables targeted explanations. |

be interpreted as initial cross-platform evidence rather than proof of universal generalizability. Independent validation on enterprise and self-hosted platforms is needed to assess taxonomy applicability in those contexts.

**Internal validity limitations:** Feature selection was constrained to metrics available in TravisTorrent and GHALogs schemas. Alternative features capturing developer expertise [58], team dynamics, or organizational context could enhance prediction but were unavailable. The Random Forest hyperparameters (100 trees, max depth 10) represent validated defaults; alternative optimization strategies might identify superior configurations for specific contexts.

**Temporal validity:** The 10-year span addresses ecosystem evolution concerns, but the 2018–2022 gap (encompassing Docker ubiquity, Kubernetes adoption, and GitHub Actions launch) means models trained on 2013–2016 data may require retraining for current practices. Organizations should monitor concept drift via Population Stability Index and retrain when performance degrades.

**Differential robustness:** The structural insight that project maturity dominates code metrics likely generalizes across contexts, reflecting enduring organizational dynamics rather than transient technical characteristics. Quantitative performance estimates (82.73% accuracy) require validation on recent data. The leakage prevention methodology represents the most generalizable contribution, applicable across software engineering prediction domains regardless of dataset or platform.

## Future research directions

- **Fine-grained failure classification**: Extending binary prediction to multi-class prediction distinguishing compilation errors, test failures, infrastructure timeouts, and dependency issues would enable targeted interventions via build log parsing or manually annotated datasets.

- **Causal inference**: Applying Pearl's do-calculus [52], instrumental variables, or difference-in-differences methods could transform correlational feature importance into actionable counterfactual recommendations (e.g., quantifying the expected build success improvement from increasing test density).

- **Transfer learning**: Pre-training foundation models on large-scale public datasets [59,60] then fine-tuning on 100–1,000 organization-specific builds could enable zero-shot prediction for new projects. Federated learning [61] would address privacy concerns across organizations.

- **Multi-objective build scheduling**: Integrating prediction models with CI/CD orchestration systems to balance build queue times, failure detection rates, and infrastructure costs through Pareto-optimal scheduling policies.

- **Explainable AI**: SHAP values and LIME could generate build-specific risk explanations (e.g., quantifying each feature's contribution to failure probability), building developer trust and enabling targeted corrective actions.

## Conclusions

This study developed a three-type temporal data leakage taxonomy and applied it to CI/CD build prediction across 175,706 builds on two platforms spanning 10 years. The taxonomy addresses a systemic methodological problem in software engineering prediction research: studies reporting 95–99% accuracy [5] include features unavailable at prediction time, producing results that are scientifically invalid for deployment assessment. By systematically identifying and removing 35 leaky features, the taxonomy establishes realistic prediction baselines (82.73–83.30%) that represent genuine predictive capability rather than retrospective data contamination.

The significance of these findings extends beyond build prediction. The 14.59pp divergence in leakage tax between platforms (15.07pp Travis CI versus 0.48pp GitHub Actions) reveals that temporal leakage vulnerability is platform-dependent, a finding absent from prior literature that assumed uniform leakage effects. Equally consequential, the 6.5:1 dominance of project

context over code metrics (49.8% versus 7.7% importance) challenges the code-centric paradigm prevalent in software engineering research, suggesting that organizational factors (developer retention, process maturity, testing infrastructure) are stronger predictors of software quality than code characteristics alone. The taxonomy generalizes to defect prediction, test selection, and code review automation, providing detection rules applicable across software engineering prediction domains.

These results are limited to open-source projects on two platforms across four languages (Java, Ruby, Python, JavaScript) with binary classification only, and a 2018–2022 validation gap exists. Extension to enterprise platforms (Jenkins, GitLab CI), multi-class failure prediction, and causal inference methods represents the most promising future directions. Complete replication packages are publicly available (DOI: 10.5281/zenodo.17745286).

## Acknowledgments

The authors thank Moritz Beller, Georgios Gousios, and Andy Zaidman for creating and maintaining the TravisTorrent dataset, and Malinda Dilhara, Abhishek Sharma, and Danny Dig for developing and releasing the GHALogs dataset. Both open-access datasets were instrumental in enabling the cross-platform validation study presented in this work. The open-source community is also acknowledged for maintaining the CI/CD infrastructure (Travis CI and GitHub Actions) that generated the underlying build execution data.

### Use of AI tools

During the preparation of this manuscript, the authors used Claude Code (Anthropic, Claude Opus 4.5) for assistance with text drafting and editing. Specifically, the AI tool was used to: (1) improve clarity and readability of technical explanations and (2) ensure consistent terminology throughout the manuscript.

The authors critically reviewed and verified all AI-generated content against the original experimental data and analysis code. All scientific interpretations, hypotheses, conclusions and claims of contribution represent the authors' own intellectual work. The authors take full responsibility for the accuracy and validity of all content. No AI tools were used to generate, analyze or manipulate the research data or figures.

## Author contributions

**Conceptualization:** Lalit Narayan Mishra, Amit Rangari.

**Data curation:** Lalit Narayan Mishra, Amit Rangari.

**Formal analysis:** Lalit Narayan Mishra, Amit Rangari.

**Methodology:** Lalit Narayan Mishra, Sandesh Nagrare.

**Project administration:** Sandesh Nagrare.

**Software:** Lalit Narayan Mishra, Amit Rangari, Sandesh Nagrare.

**Supervision:** Amit Rangari, Saroj Kumar Nayak.

**Validation:** Lalit Narayan Mishra, Sandesh Nagrare, Saroj Kumar Nayak.

**Visualization:** Amit Rangari, Sandesh Nagrare.

**Writing – original draft:** Lalit Narayan Mishra, Amit Rangari.

**Writing – review & editing:** Lalit Narayan Mishra, Amit Rangari, Sandesh Nagrare, Saroj Kumar Nayak.

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
