## [Decision Letter · Decision Letter 0]

5 Feb 2026

PONE-D-25-66966A taxonomy for detecting and preventing temporal data leakage in machine learning-based build prediction: A dual-platform empirical validationPLOS One

Dear Dr. Mishra,

Thank you for submitting your manuscript to PLOS ONE. After careful consideration, we feel that it has merit but does not fully meet PLOS ONE’s publication criteria as it currently stands. Therefore, we invite you to submit a revised version of the manuscript that addresses the points raised during the review process.

We look forward to receiving your revised manuscript.

Kind regards,

Syed Hamid Hussain Madni

Academic Editor

PLOS One

Journal Requirements:

4. We notice that your supplementary figures are uploaded with the file type 'Figure'. Please amend the file type to 'Supporting Information'. Please ensure that each Supporting Information file has a legend listed in the manuscript after the references list.

5. Please upload a copy of Supporting Information Figure/Table/etc. Supporting Information which you refer to in your text on page 44 and 45.

Reviewers' comments:

Reviewer's Responses to Questions

**Comments to the Author**

1. Is the manuscript technically sound, and do the data support the conclusions?

Reviewer #1: Yes

Reviewer #2: Partly

2. Has the statistical analysis been performed appropriately and rigorously? 

Reviewer #1: Yes

Reviewer #2: No

3. Have the authors made all data underlying the findings in their manuscript fully available?

Reviewer #1: Yes

Reviewer #2: Yes

4. Is the manuscript presented in an intelligible fashion and written in standard English?

Reviewer #1: Yes

Reviewer #2: No

5. Review Comments to the Author

Reviewer #1: 1. Revise the abstract so that it is presented as a single, coherent paragraph.

2. The introduction section is overly lengthy and should be revised to more clearly highlight the contributions of previous studies in addressing the identified problem. Clearly articulate the existing gap or prevailing challenge, then introduce the proposed methodology with a logical justification explaining why it is suitable for addressing the problem.

3. Explicit research questions are not strictly necessary in a research article. You may remove this section and instead clearly state the main contributions of the paper.

4. Revise the related work section to be more concise, focusing primarily on recent and closely related studies published in reputable journals.

5. The presentation of several sections resembles a tutorial rather than a research article. Consider replacing lengthy textual explanations with figures, flowcharts, and tables where appropriate to improve clarity and conciseness.

6. There is an overreliance on accuracy as the primary performance metric despite the presence of class imbalance, particularly in the GHALogs dataset. This limits the operational interpretability of the results, as practical CI/CD deployment decisions often depend more on precision–recall trade-offs than on raw accuracy. Incorporate cost-sensitive analysis, precision–recall curves, and scenario-based error cost evaluations.

7. Although strong claims of generalizability are made, the evaluation is limited to only two CI/CD platforms. No validation is provided for self-hosted CI systems such as Jenkins, enterprise CI platforms such as GitLab CI and Azure DevOps, or monorepo-heavy environments.

8. The conclusion section currently reads more like a summary. Rewrite it to concisely restate the proposed approach and interpret the results, highlighting the significance of the findings, the impact of the study, and potential directions for future research.

Reviewer #2: be concise and to the point, there are lots of repetitive information, paragrapghs starts just at the end of tables caption, its more looks like a thesis or book chapter instead of paper, strong need of work to be presented in a research paper style,lots of use of "we". lots of improvements required, i have highlighted many.

6. PLOS authors have the option to publish the peer review history of their article (what does this mean?). If published, this will include your full peer review and any attached files.

Reviewer #1: **Yes:** Ayuba John

Reviewer #2: No

---

## [Author Response · Author response to Decision Letter 1]

25 Feb 2026

Please see the attached "response_to_reviewers.pdf" for our detailed point-by-point response to all reviewer comments. A tracked-changes PDF highlighting all modifications is also included.

---

## [Decision Letter · Decision Letter 1]

21 Apr 2026

A taxonomy for detecting and preventing temporal data leakage in machine learning-based build prediction: A dual-platform empirical validation

PONE-D-25-66966R1

Dear Dr. Mishra,

We’re pleased to inform you that your manuscript has been judged scientifically suitable for publication and will be formally accepted for publication once it meets all outstanding technical requirements.

Kind regards,

Elochukwu Ukwandu, PhD

Academic Editor

PLOS One

Additional Editor Comments (optional):

Reviewers' comments:

Reviewer's Responses to Questions

**Comments to the Author**

1. If the authors have adequately addressed your comments raised in a previous round of review and you feel that this manuscript is now acceptable for publication, you may indicate that here to bypass the “Comments to the Author” section, enter your conflict of interest statement in the “Confidential to Editor” section, and submit your "Accept" recommendation.

Reviewer #1: All comments have been addressed

Reviewer #2: All comments have been addressed

2. Is the manuscript technically sound, and do the data support the conclusions?

Reviewer #1: Yes

Reviewer #2: Partly

3. Has the statistical analysis been performed appropriately and rigorously? 

Reviewer #1: Yes

Reviewer #2: Yes

4. Have the authors made all data underlying the findings in their manuscript fully available?

Reviewer #1: Yes

Reviewer #2: Yes

5. Is the manuscript presented in an intelligible fashion and written in standard English?

Reviewer #1: Yes

Reviewer #2: Yes

6. Review Comments to the Author

Reviewer #1: After reviewing the authors’ responses and the revised manuscript, I am impressed with the meticulous way they addressed all the reviewers’ comments to a commendable extent. I have no doubt that the manuscript has been significantly improved and is now suitable for publication in this esteemed journal for the benefit of its readers.

Reviewer #2: (No Response)

7. PLOS authors have the option to publish the peer review history of their article (what does this mean?). If published, this will include your full peer review and any attached files.

Reviewer #1: **Yes:** Ayuba John

Reviewer #2: **Yes:** R.Marriam

---

## [Editor Report · Acceptance letter]

PONE-D-25-66966R1

PLOS One

Dear Dr. Mishra,

I'm pleased to inform you that your manuscript has been deemed suitable for publication in PLOS One. Congratulations! Your manuscript is now being handed over to our production team.

Kind regards,

on behalf of

Dr. Elochukwu Ukwandu

%CORR_ED_EDITOR_ROLE%

PLOS One